# The association between all-cause mortality and HIV acquisition risk groups in the United States, 2001–2014

Fatma M. Shebl[1,2]*, Yiqi Qian[1], Julia H. A. Foote[1,2], Nattanicha Wattananimitgul[1], Krishna P. Reddy[1,2,3], Anne M. Neilan[1,2,4,5], Andrea L. Ciaranello[1,2,5,6], Elena Losina[6,7,8], Kenneth A. Freedberg[1,2,5,6,9,10], Emily P. Hyle[1,2,5,6]

1 Medical Practice Evaluation Center, Massachusetts General Hospital, Boston, Massachusetts, Unite States of America, 2 Harvard Medical School, Boston, Massachusetts, United States of America, 3 Division of Pulmonary and Critical Care Medicine, Massachusetts General Hospital, Boston, Massachusetts, United States of America, 4 Division of General Academic Pediatrics, Massachusetts General Hospital, Boston, Massachusetts, United States of America, 5 Division of Infectious Diseases, Massachusetts General Hospital, Boston, Massachusetts, United States of America, 6 Harvard University Center for AIDS Research (CFAR), Cambridge, Massachusetts, United States of America, 7 Department of Orthopedic Surgery, Brigham and Women's Hospital, Boston, Massachusetts, United States of America, 8 Department of Biostatistics, Boston University School of Public Health, Boston, Massachusetts, United States of America, 9 Division of General Internal Medicine, Massachusetts General Hospital, Boston, Massachusetts, United States of America, 10 Department of Health Policy and Management, Harvard T.H. Chan School of Public Health, Boston, Massachusetts, United States of America

* fshebl@gmail.com

**Data Availability Statement:** All data from National Health and Nutrition Examination Survey and Linked Mortality files are available from the CDC

## Abstract

### Objective

To investigate associations between all-cause mortality and human immunodeficiency virus (HIV) acquisition risk groups among people without HIV in the United States.

### Methods

We used data from 23,657 (NHANES) participants (2001–2014) and the Linked Mortality File to classify individuals without known HIV into HIV acquisition risk groups: people who ever injected drugs (ever-PWID); men who have sex with men (MSM); heterosexually active people at increased risk for HIV (HIH), using low income as a proxy for increased risk. We used Cox proportional hazards models to estimate adjusted and unadjusted all-cause mortality hazard ratios (HR) with 95% confidence intervals (CI).

### Results

Compared with sex-specific heterosexually active people at average risk for HIV (HAH), the adjusted HR (95% CI) were: male ever-PWID 1.67 (1.14, 2.46), female ever-PWID 3.50 (2.04, 6.01), MSM 1.51 (1.00, 2.27), male HIH 1.68 (1.04, 2.06), female HIH 2.35 (1.87, 2.95), and male ever-PWID 1.67 (1.14, 2.46).

database at www.cdc.gov/nchs/nhanes/index.htm and www.cdc.gov/nchs/data-linkage/mortality.htm.

**Funding:** This work was supported by the National Institute of Allergy and Infectious Disease (R01AI042006 [KAF]) and R01AI093269 [KPR]), the Eunice Kennedy Shriver National Institute of Child Health and Human Development (K08HD094638 [AMN]), and the National Heart, Lung, and Blood Institute (K01HL123349 [EPH]) of the National Institutes of Health (https://www.nih.gov/), the National Institute on Aging (R01AG069575 [EPH]), and the Steve and Deborah Gorlin MGH Research Scholars Award (KAF), the MGH Jerome and Celia Reich Endowed Scholar Award (EPH), and the MGH Scholar Award in Population and Health Care Research (ALC) of the MGH Executive Committee on Research (https://ecor.mgh.harvard.edu/). The funders had no role in study design, data collection and analysis, decision to publish, or preparation of the manuscript.

**Competing interests:** The authors have declared that no competing interests exist.

## Conclusions

Most people at increased risk for HIV in the US experience higher all-cause mortality than people at average risk. Strategies addressing social determinants that increase HIV risk should be incorporated into HIV prevention and other health promotion programs.

## Introduction

With antiretroviral therapy, HIV-related mortality has declined markedly, leading to improved life expectancy for people with HIV (PWH) [1, 2]. However, PWH often experience greater mortality compared with the general population due to an increased risk of non-HIV-related diseases and health conditions, such as cancer, liver disease, cardiovascular disease, hepatitis C virus (HCV), tobacco use, and substance use disorder [3–5]. Although some of the increased non-HIV-related mortality among PWH may be due directly to HIV infection and chronic inflammation [6], there is also likely to be a contribution from health conditions and social determinants of health that are not directly related to HIV status. To better understand the extent to which this excess mortality is due to HIV-related versus non-HIV-related causes, we focused on people without HIV who met the criteria for HIV acquisition risk groups as defined by the Centers for Disease Control and Prevention (CDC) [7].

The CDC-defined HIV acquisition risk groups that experience the highest burden of HIV in the United States include people who ever injected drugs (ever-PWID), men who have sex with men (MSM), and heterosexually active people at increased risk for HIV (HIH), which has been associated with low income [7–9]. Previously published studies have demonstrated a range of findings regarding excess mortality among people in HIV acquisition risk groups, depending on the definition and comparator groups [8, 10–13].

To quantify excess all-cause mortality among these key populations without HIV, we estimated mortality rates and compared age-adjusted, sex-stratified mortality with people at average risk for HIV acquisition in the United States to assess for associations between all-cause mortality and HIV acquisition risk groups.

## Methods

### Study population and data sources

We used publicly available data from the 2001 to 2014 National Health and Nutrition Examination Survey (NHANES) [14] and the Linked Mortality File (LMF) [15], which has been updated with National Death Index (NDI) mortality follow-up data through December 31, 2019. The LMF includes death dates for those who have died [15], and NHANES is a multi-stage study that summarizes the health status of noninstitutionalized civilians residing in the United States, including interviews and physical examinations [14].

To define the study population, we first identified the population eligible for the mortality analysis (people 18 years or older with linked mortality data), to whom the questionnaires that allow classification into one of the HIV acquisition risk groups were administered and had no missing data (N = 23,657). Using HIV antibody test information from NHANES laboratory data, we restricted the population to participants without an HIV diagnosis by excluding individuals with a reactive HIV antibody test result (N = 101). We also excluded participants who died without an underlying cause of death listed in the linked mortality files. Among participants with the cause of death listed as "other," we excluded individuals without a documented, non-reactive HIV antibody test result (N = 152) because HIV-associated causes of death could

be included in this "other" category. We also excluded women who had sex with women without a history of injection drug use (N = 1,008) because they are not defined as an HIV acquisition risk group by the CDC [7] and are not "heterosexually active at low risk for HIV acquisition." After applying all the exclusion criteria, 22,396 adults met the study-defined inclusion criteria (S1 Fig in S4 Appendix).

## Ethical approval

The research ethics review board at the National Center for Health Statistics (NCHS), approved the study protocol [14]. This project was reviewed by the institutional review board at Mass General Brigham and was deemed as "not human subjects research".

## Outcome

**All-cause mortality.** We ascertained mortality statuses, dates, and causes of death from the public-use National Death Index (NDI)-linked data of survey participants followed through December 31, 2019 [15]. Follow-up time included the time from the NHANES interview date until death (for those who died) or December 31, 2019 (for living participants) [15].

To estimate mortality by age-at-risk, we conducted Lexis expansion to stratify follow-up by intervals of one year, converting the one observation per person to one observation for each time interval (one year of age) per person so that each participant contributes to each age stratum until death or censoring [16].

## Exposures

**People who ever injected drugs (ever-PWID).** We categorized participants as "ever-PWID" if they self-reported ever using a needle to inject illegal or street drugs based on their answers to drug use questionnaires available for NHANES participants aged 20–59 years (S1 Appendix). All other participants were classified as "never-PWID" or unknown PWID status based on their self-response to the PWID questions.

**Men who have sex with men (MSM).** We classified male participants as MSM if they reported that they were gay, bisexual, or had at least one sexual experience with a man during their lifetime; we categorized all other male participants as heterosexually active if they responded to the sexual behavior questions and did not meet the MSM criteria (S1 Appendix).

**Heterosexually active people at increased risk for HIV (HIH).** Because the risk profiles of people's sexual partners are often unknown, it is challenging to apply and operationalize the CDC definition of "high-risk heterosexual contact" (i.e., persons who have ever had heterosexual contact with a person known to have, or to be at high risk for, HIV infection) [7]. The National HIV Behavior Surveillance (NHBS) uses income as a proxy for being at increased risk for HIV acquisition because HIV prevalence is markedly higher among heterosexually active people with lower incomes compared with higher incomes in the United States. The NHBS, therefore, recruits heterosexually active male and female participants from census tracts areas in which at least 25% of residents live below the U.S. Census Bureau's poverty threshold [17]. Among male and female participants categorized as heterosexually active, we used the poverty income ratio (PIR) as an indicator for SES [14, 18]. We defined PIR as the ratio of the midpoint of the observed family income category to the poverty guidelines from the Department of Health and Human Services in each calendar year to adjust for inflation across NHANES survey years [14, 18]. We categorized heterosexually active participants as HIH if their PIR was <1.0, which is the official federal poverty line [18]. The remainder of heterosexually active participants who had PIR ≥1.0 were classified as at average risk for HIV acquisition (HAH) [18].

**HIV acquisition risk group categorization.** Because some participants met the criteria for more than one HIV acquisition risk group (e.g., MSM who have ever used injection drugs), we used the CDC hierarchy of risk factor assignment to categorize participants into one of the five mutually exclusive HIV acquisition risk groups: 1) male ever-PWID (including MSM and male HIH who ever injected drugs), 2) female ever-PWID (including female HIH who ever injected drugs), 3) MSM (who have never injected drugs), 4) male HIH (who are not MSM and have never injected drugs), and 5) female HIH (who have never injected drugs). The remaining participants met the criteria for heterosexually active persons at average risk for HIV acquisition (HAH), stratified by self-reported gender as used in NHANES questionnaires. In sensitivity analysis, we categorized participants into multiple HIV acquisition risk groups (e.g., male HIH who inject drugs contributes to both male ever-PWID and male HIH).

**Covariates of interest.** To examine mortality rates among individuals in the HIV acquisition risk groups, we assessed an *a priori* list of potential effect modifiers and confounders based on the published literature, including age, race/ethnicity, education, health insurance, health status, at-risk alcohol use, tobacco use, body mass index, condom use, lifetime sexual partners, and history of sexually transmitted infections [11, 12, 19–22]. These effect modifiers and cofounders are further defined in S1 Appendix.

## Statistical analysis

We used SAS 9.4 to perform the statistical analyses (Cary, NC, USA). We used design weights, strata, and primary sampling units in all analyses, as suggested by the National Center for Health Statistics (NCHS) [23]. To correct for possible estimation bias resulting from restricting the analysis to the eligible sample, we inflated the sampling weights [24]. We calculated unweighted counts, weighted percentages, and weighted and unweighted rates (the number of deaths divided by the follow-up time). For bivariate associations, we used the Rao-Scott F adjusted chi-square statistic to assess the relationship between covariates and the primary exposure (HIV acquisition risk groups) (i.e. to assess whether the distribution of the proportions of HIV acquisition risk groups is independent of the covariate) [25]. We assessed the relationship between covariates and the outcome of interest (all-cause mortality) using Cox proportional hazards regression models, accounting for the complex survey design [25]. Cox proportional hazards regression model is a semiparametric model; therefore, no assumption of the baseline hazard function is needed. Results of the bivariate associations were used as the first step in assessing the *a priori* list of potential confounders.

We also examined effect modification and confounding to identify covariates other than the HIV acquisition risk group to be included in the final, most parsimonious multivariable regression models. We stratified risk groups by gender for two reasons, 1) there was a statistically significant interaction between the four-level risk group and gender on all-cause mortality, 2) it is more appropriate to compare women to women and men to men to account for unmeasured confounders that are related to self-reported gender, and the difference in mortality rates between men and women. We then assessed whether "age at risk" is a potential effect modifier by including HIV acquisition risk group, age and the interaction term between age and HIV acquisition risk group in Cox proportional hazards model. Age would be considered as an effect modifier, if the p value of the interaction term was less than the alpha level of .05.

Subsequently, we assessed confounding due to covariates that were not effect modifiers. For each potential confounder, we calculated the difference between crude and adjusted regression coefficient *betas*, which we compared with the adjusted regression coefficient *beta*. We considered the covariate to be a potential confounder if the ratio of the difference in the *betas* to the

adjusted *beta* was greater than 10%. Subsequently, only variables with an alpha level < .05 were retained in the final, most parsimonious multivariable regression models.

Additionally, we assessed the Cox proportional hazard assumption using two standard approaches: 1) to visually assess the parallelisms of the lines of the graph of the log of -log survival versus the log of survival time (log-negative log plot), and 2) to include and assess the significance of the time-dependent variable in the model. If the lines were parallel and the time-dependent variable was not significant, then we considered the proportional hazard assumption to not be violated.

## Results

At the time of the survey interview, the mean age (standard error [SE]) of participants was 36.5 (0.22) years; 11,526 (52.2%) were male, and 10,870 (47.8%) were female. Participants self-identified as non-Hispanic White (10,004 [69.2%]), Black/African American (4,789 [11.0%]), Hispanic (5,814 [13.8%]), and Other (1,789 [6.0%]).

### HIV acquisition risk groups

Of the 22,396 participants who met the study-defined inclusion criteria, 509 (2.3%) participants were ever-PWID (male, 340 [68.2%]; female, 169 [31.8%]); 565 (2.8%) participants were MSM; and 4,558 (13.5%) heterosexually active persons met the criteria for low income and were classified as HIH (male, 2,133 [47.0%]; female, 2,425 [53.0%]; S1 Fig in S4 Appendix). The remaining 16,764 (81.5%) participants met the criteria for being at average risk for HIV acquisition (male, 8,488 [51.0%]; female, 8,276 [49.0%]). The associations between demographic characteristics and HIV acquisition risk groups are presented in Fig 1, S2 Appendix and S1 Table in S3 Appendix.

### Mortality

During the follow-up period, 22,396 participants contributed 249,326 person-years (PY), and 1,139 deaths were reported (S2 Table in S3 Appendix). The weighted all-cause mortality rate was 407.1 deaths/100,000 PY. The unweighted all-cause mortality rates and the associations of mortality with population characteristics are also provided in S2 Table (S3 Appendix).

**Mortality association with HIV acquisition risk groups.**    Among participants categorized into the mutually exclusive HIV acquisition risk groups, the weighted all-cause mortality rate was highest among female ever-PWID (1,289.8 deaths/100,000 PY), followed by male ever-PWID (1,105.1 deaths/100,000 PY, Table 1). Mortality rates were similar among MSM (626.0 deaths/100,000 PY) and male HIH (631.7 deaths/100,000 PY), with lower mortality among female HIH (552.7 deaths/100,000 PY). When combining all people at high risk for HIV acquisition (i.e., male and female ever-PWID, MSM, and male and female HIH), the weighted all-cause mortality rate was 666.6 deaths/100,000 PY. Persons at average risk for HIV acquisition had significantly lower mortality rates (males: 443.5 deaths/100,000 PY; females: 256.2 deaths/100,000 PY; and males and females combined: 351.1 deaths/100,000 PY). We detected no significant interaction between age and HIV acquisition risk groups, $F(6,104) = 1.88$, $p = .091$ (S3 Table in S3 Appendix). S4 Table in S3 Appendix displays mortality rates when participants were categorized into multiple risk acquisition groups.

**Multivariable regression models of the association between mortality and HIV acquisition risk groups.**    We did not detect a violation of the proportional hazard assumption. After adjusting for confounders including age, race/ethnicity, education, smoking status, at-risk alcohol consumption, and health status, the adjusted all-cause mortality hazard ratio (aHR, 95% CI) was significantly higher among female ever-PWID (aHR, 3.50 [95% CI 2.04, 6.01]),

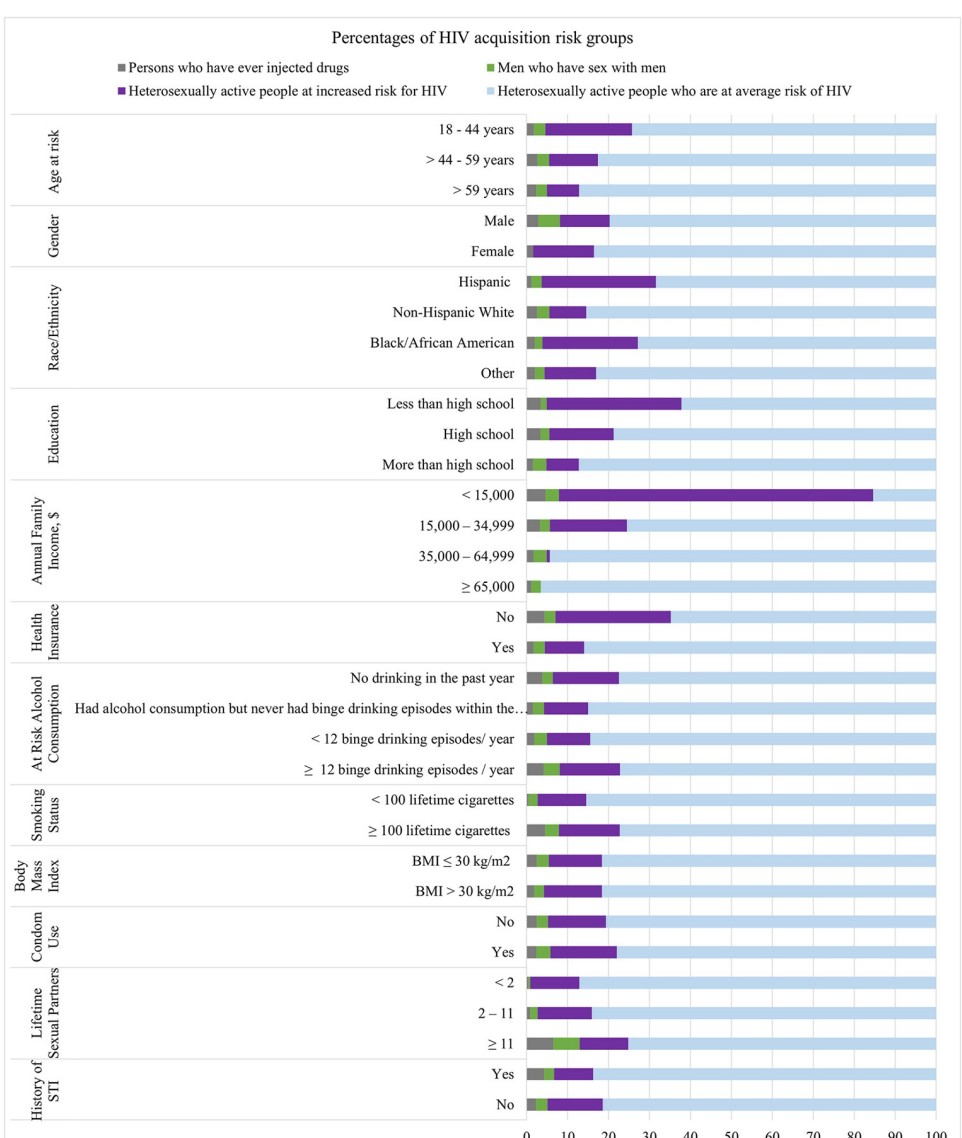

**Fig 1. Weighted percentages of risk groups by participants' characteristics.** We display the weighted percentages of the four mutually exclusive groups, **PWID**, people who ever injected drugs (grey); **MSM**, men who have sex with men (green); **HIH**, heterosexually active people at increased risk for HIV (purple); **HAH**, heterosexually active people at average risk for HIV (blue), within each of the population characteristics. **BMI**, body mass index; **STI**, sexually transmitted infection.

male ever-PWID (aHR, 1.67 [95% CI 1.14, 2.46]), MSM (aHR, 1.51 [95% CI 1.00, 2.27]), male HIH (aHR, 1.68 [95% CI 1.36, 2.06]), and female HIH (aHR, 2.35 [95% CI 1.87, 2.95]) compared with male and female HAH, respectively (Table 2).

The adjusted hazard ratio estimates did not change substantially when participants with unknown HIV acquisition risk categories were included in the Cox model (S5 Table in S3 Appendix).

## Discussion

In a representative sample of the noninstitutionalized US adult population not diagnosed with HIV, we found that the age-adjusted, sex-stratified, all-cause mortality among people who self-

**Table 1. All-cause mortality rates by mutually exclusive HIV acquisition risk groups\*.**

| Covariate | Participants, N (%)[a] | Deaths N | Length of follow-up years[b] | Unweighted all-cause mortality rate per 100,000 PY | Weighted all-cause mortality rate per 100,000 PY[c,d] | p-value[e] |
|---|---|---|---|---|---|---|
| **HIV Acquisition Risk Groups** | | | | | | <0.0001 |
| Male HAH | 8,488 (41.6) | 468 | 94,592 | 494.8 | 443.5 | |
| Female HAH | 8,276 (39.9) | 260 | 94,883 | 274.0 | 256.2 | |
| Male ever-PWID | 340 (1.5) | 50 | 3,610 | 1,385.0 | 1,105.1 | |
| Female ever-PWID | 169 (0.7) | 26 | 1,780 | 1,460.3 | 1,289.8 | |
| MSM | 565 (2.8) | 37 | 6,039 | 612.7 | 626.0 | |
| Male HIH | 2,133 (6.3) | 170 | 22,377 | 759.7 | 631.7 | |
| Female HIH | 2,425 (7.1) | 128 | 26,044 | 491.5 | 552.7 | |
| **Increased HIV Acquisition Risk** | | | | | | <0.0001 |
| All high-risk groups HAH | 16,764 (81.5) | 728 | 189,476 | 384.2 | 351.1 | |
| Combined [f] | 5,632 (18.5) | 411 | 59,851 | 686.7 | 666.6 | |

**PY,** person-year; **MSM,** men who have sex with men; **PWID,** people who ever injected drugs; **HIH,** heterosexually active people at increased risk for HIV; **HAH,** heterosexually active people at average risk for HIV.

[a] Weighted percentage

[b] Unweighted, rounded to one full year

[c] Adjusted for the design weights, strata, and primary sampling units

[d] The time of death or the time of exit from the study

[e] P-value from Cox proportional hazards regression model

[f] Included ever-PWID, MSM, HIH.

reported CDC-defined risks for HIV acquisition was significantly higher compared with heterosexually active people at average risk for HIV. Although risk factors for mortality among adults with HIV have previously been well-described [5, 26, 27], few studies have examined the mortality rates for people without diagnosed HIV who can be categorized into one of the CDC-defined HIV acquisition risk groups [11–13, 28, 29].

**Table 2. Unadjusted and adjusted association of mutually exclusive HIV acquisition risk group and all-cause mortality.**

| HIV Acquisition Risk Group | Unadjusted [a,b] | Adjusted [a,b,c] |
|---|---|---|
| | All-cause mortality HR (95% CI) | All-cause mortality HR (95% CI) |
| HAH [e] | Ref. | Ref. |
| Male PWID | 2.49 (1.72, 3.61) | 1.67 (1.14, 2.46) |
| Female PWID | 5.04 (3.03, 8.38) | 3.50 (2.04, 6.01) |
| MSM | 1.41 (0.94, 2.12) | 1.51 (1.00, 2.27) |
| Male HIH [d] | 1.42 (1.15, 1.76) | 1.68 (1.36, 2.06) |
| Female HIH [d] | 2.16 (1.75, 2.67) | 2.35 (1.87, 2.95) |

**HR,** hazard ratio; **CI,** confidence interval; **MSM,** men who have sex with men; **PWID,** people who inject drugs; **HIH,** heterosexually active people at increased risk for HIV; **HAH,** heterosexually active people at average risk for HIV

[a] Cox proportional hazards regression model.

[b] Adjusted for the design weights, strata, cluster, and primary sampling units.

[c] Adjusted for age, race/ethnicity, education, smoking status, at-risk alcohol consumption, and health status.

[d] Low socioeconomic status is used as a proxy for heterosexually active people at increased risk for HIV.

[e] Sex-specific reference categories.

We found the highest unadjusted crude all-cause mortality rate among people at risk for HIV among people who ever used injection drugs, at more than double the rate for heterosexually active people at average risk for HIV. Other studies have found similarly increased crude all-cause mortality rates among people who ever used injection drugs compared with the general population or with people who have never used injection drugs, due to ongoing substance use, infections other than HIV, drug overdose, sudden cardiac death, or serious mental illness [8, 30–33]. Our results also support previously reported gender differences in mortality among people who ever used injection drugs. Although crude mortality rates are often higher among men who use injection drugs than women [30–32], adjusted mortality relative risks are typically higher among women [8, 30–32] as our analysis also shows. The attenuation of the magnitude of difference in mortality between male injection drug users and heterosexually active males at average risk for HIV after adjusting for confounders indicates that the excess mortality could be attributed to the health risk behaviors such as smoking and alcohol consumption, as well as racial disparities.

The magnitude of increased mortality among people who ever used injection drugs has been greater in other analyses than that reported in our analysis; standardized mortality ratios (SMR) as high as 14.68 (95% CI: 13.01–16.35) [30] and 11.09 (95% CI: 6.68–18.39) have been previously reported [33]. The current analysis may have found less excess mortality because participants in NHANES may underrepresent people who are either actively using injection drugs or are at risk of being institutionalized, incarcerated, or experiencing marginal housing. Therefore, estimated excess mortality among NHANES participants is likely to be an underestimate compared with the entire population of people ever using injection drugs. Additionally, HCV-related mortality was greater in the years prior to effective curative treatment and may have been included in other published estimates. In some published analyses, PWID with HIV were also included in mortality estimates, which would further increase mortality [8, 30]. With rising rates of overdose in the setting of fentanyl contamination [34], excess mortality among people who use injection drugs is likely to increase further; systematic approaches to support access to treatment are essential to reduce mortality.

Our results also suggest that MSM without diagnosed HIV infection may experience excess mortality compared with heterosexually active persons at average risk for HIV acquisition, after adjustment for confounders. Previously published studies have shown no increase in all-cause mortality among MSM without HIV compared with heterosexually active males [22, 28], although other studies from NHANES showed a trend similar to our findings [11, 12]. We found an increased risk of all-cause mortality among self-identified MSM compared with heterosexually active men with incomes above the poverty line, as a proxy for average risk for HIV acquisition. Importantly, this analysis used heterosexually active men with incomes above the poverty line as the comparator group and excluded individuals who inject drugs, whereas other studies used all heterosexually active men, which could bias the results towards the null given the inclusion of heterosexually active men with low income or who inject drugs in the heterosexually active men comparator group [11, 12, 22, 28]. The observed excess all-cause mortality could be due to previously reported increased risk of suicide, tobacco use, at-risk alcohol use, non-injection substance use, mental health disorders, and other adverse social determinants of health among MSM [29, 35, 36]. Our findings highlight the importance of an improved understanding of risk factors for excess mortality among MSM independent of HIV status.

Age-adjusted, sex-stratified all-cause mortality rates were higher among heterosexually active adults at increased risk for HIV acquisition compared with heterosexually active adults at average risk, as defined by the poverty-to-income ratio. Our results are consistent with other studies regarding the magnitude of the association between low income or socioeconomic

status and all-cause mortality. A multi-cohort, international study of approximately 1.7 million participants from seven high income countries reported a 30–40% excess mortality among people with low socioeconomic status compared with people with high socioeconomic status [37]. A recently published analysis underscored that increasing numbers of disadvantageous social determinants of health were associated with all-cause mortality in the United States with a significant linear trend; among people with six or more disadvantageous social determinants of health, all-cause mortality was increased more than 7-fold [38]. Our findings highlight that heterosexually active people at increased risk for HIV acquisition also experience increased mortality, which further highlights the need to address social determinants of health and improve health outcomes among people with low income who are at risk for HIV acquisition.

This study has several limitations. First, data regarding sexual behaviors and injection drug use were self-reported and only available at the initial NHANES interview. However, if participants' risk behaviors changed during follow-up, the comparative effect on mortality would be further reduced. Second, although misclassification of HIV status is possible given that NHANES public-use data does not include details regarding HIV status among all participants, we excluded any possible HIV-related mortality and missing data that are likely to be missing at random. Although we adjusted for multiple confounders, including smoking, alcohol use, health status, and race/ethnicity, residual confounding may persist due to missing data, measurement error, unreported confounders, or incomplete self-report especially because NHANES does not include data regarding time-varying covariates [39]. Given small numbers of outcomes, we were not able to perform the analyses stratified by race. The use of change in estimate approach to identify confounders has been criticized [40]. However, since it is advised not to control for variables that are not confounders to avoid introducing "unnecessary adjustment" bias, [41] we used the change in estimate criterion as a complementary method to evaluate a list of *a priori* potential confounders. In identifying the list of *a priori* potential confounders, we followed the recommendations of VanderWeele 2019, [40] by selecting variables that could be causes of the exposure, outcome, or both, but excluding colliders, mediators and instrumental variables. Although the dataset is large, we observed a sparse data problem because the outcome is relatively rare, which precluded us from including all the potential confounders in the model [42]. Therefore, we elected not to adjust except for variables associated with both outcome and exposure that meaningfully changed the estimate as determined by the change in the estimate approach. Last, our results might not be generalizable to all people at increased risk of HIV acquisition given that people with active injection drug use, serious mental illness, or disadvantageous social determinants of health may be less likely to participate in NHANES.

This study highlights that people at increased risk for HIV acquisition in the US have greater all-cause mortality rates compared with people at average risk. Although the gap in life expectancy between people with HIV and people without HIV is decreasing with effective antiretroviral therapy [1, 2], more attention must be focused on improving health outcomes that are not directly related to HIV status but affect ever-PWID, MSM, and heterosexually active people with low income. Clinical and public health efforts should be focused on improving access to care and services that address risks and social determinants of health in people at increased risk for HIV.

## Supporting information

**S1 Appendix. Supplemental methods.**
(DOCX)

**S2 Appendix. Supplemental results.**
(DOCX)

**S3 Appendix. Supplemental tables.**
(DOCX)

**S4 Appendix. Supplemental figures.**
(DOCX)

## Acknowledgments

We thank Ms. Stephanie Horsfall, Mr. Kyu-Young Kevin Chi, Mr. Munashe Machoko and Ms. Dina Ashour for their assistance with manuscript preparation and Ms. Michaelle Destinoble for her administrative and organizational support.

## Author Contributions

**Conceptualization:** Fatma M. Shebl, Emily P. Hyle.

**Data curation:** Fatma M. Shebl, Yiqi Qian.

**Formal analysis:** Fatma M. Shebl, Yiqi Qian.

**Funding acquisition:** Krishna P. Reddy, Anne M. Neilan, Andrea L. Ciaranello, Kenneth A. Freedberg.

**Supervision:** Fatma M. Shebl, Kenneth A. Freedberg, Emily P. Hyle.

**Writing – original draft:** Fatma M. Shebl, Emily P. Hyle.

**Writing – review & editing:** Fatma M. Shebl, Yiqi Qian, Julia H. A. Foote, Nattanicha Wattananimitgul, Krishna P. Reddy, Anne M. Neilan, Andrea L. Ciaranello, Elena Losina, Kenneth A. Freedberg, Emily P. Hyle.

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
