## [Decision Letter · Decision Letter 0]

9 Jun 2023

PONE-D-23-05219The association between all-cause mortality and HIV acquisition risk groups in the United States, 2001-2014PLOS ONE

Dear Dr. Shebl,

Thank you for submitting your manuscript to PLOS ONE. After careful consideration, we feel that it has merit but does not fully meet PLOS ONE’s publication criteria as it currently stands. Therefore, we invite you to submit a revised version of the manuscript that addresses the points raised during the review process.

We look forward to receiving your revised manuscript.

Kind regards,

Veincent Christian Pepito

Academic Editor

PLOS ONE

Journal Requirements:

Additional Editor Comments:

Dear Dr. Shebl,

Thank you very much for your submission to PLOS ONE. Here are some of my comments to your manuscript:

1. While everyone knows HIV, NHANES might be unfamiliar to those who do not live in the United States. Please write in full for first use even in the Abstract.

2. What is the rationale for excluding women who have sex with women?

3. What paper/s served as basis for your choice of covariates?

4. Why was a chi-square test done instead of a log-rank test?

5. Justify use of Cox regression over Poisson regression, considering that mortality rates are not expected to vary significantly during the time period.

5. It is unclear what effect measure modifications were assessed and why they were done. What variables were considered as effect measure modifiers?

6. There had been issues with the change in estimate criterion in more recent published literature, which you have used in your paper. What is the justification for this choice of modelling strategy, and how will your model fare if you controlled for all known covariates regardless? I think your number of deaths is sufficient even if you controlled for all covariates which you have data for.

7. How was the proportional hazards assumption assessed? Please indicate in the methods.

8. Any results about your assessment of effect modification?

9. Line 284-288: "The attenuation of the magnitude of difference in mortality between male injection drug users and heterosexually active males at average risk for HIV after adjusting for confounders indicates that the excess mortality could be attributed to the health risk behaviors such as smoking and alcohol consumption, as well as racial disparity". I thought you have adjusted for smoking, alcohol consumption and race? What made you say these statements?

10. Itemize what variables were not controlled for in the analysis.

Kindly address the comments of the reviewers as well. Thanks and I look forward to your revision.

Reviewers' comments:

Reviewer's Responses to Questions

**Comments to the Author**

1. Is the manuscript technically sound, and do the data support the conclusions?

Reviewer #1: Yes

Reviewer #2: Yes

2. Has the statistical analysis been performed appropriately and rigorously? 

Reviewer #1: Yes

Reviewer #2: Yes

3. Have the authors made all data underlying the findings in their manuscript fully available?

Reviewer #1: Yes

Reviewer #2: Yes

4. Is the manuscript presented in an intelligible fashion and written in standard English?

Reviewer #1: Yes

Reviewer #2: Yes

5. Review Comments to the Author

Reviewer #1: The authors present a well-performed and well-written analysis of the NHANES datasets to assess the inherent risks of all-cause mortality in defined groups of HIV-negative persons at higher risk of HIV acquisition. The authors are asked to address the following comments:

1. Mortality was followed-up through 2015, but obviously more data might be available in the Linked Mortality File. Please explain and defend this end date for the analysis. If more data do exist, then the analyses should be redone to account for the longer follow-up period.

2. Please defend the exclusion of women who have sex with women. If they are not at higher risk for HIV acquisition, then why not include them in the analyses?

3. Although the risk groups were stratified by sex, please clarify if there was a statistically significant interaction between risk group and sex on all-cause mortality.

4. Please use ‘sex’ or ‘sex at birth’ instead of ‘gender’ throughout, unless gender is the truly assessed variable.

Reviewer #2: The authors should consider the following.

1. Unless certain that they are PWID it should be PWUD.

2. Limitations should mention the time update variables -- eg antiretroviral therapy, viral load, nadir CD4 etc.

3. Figure 1 should be a table -- it is not clear.

4. I would like the analyses repeated by race.

5. The discussion around the social determinants of health could be expanded.

6. PLOS authors have the option to publish the peer review history of their article (what does this mean?). If published, this will include your full peer review and any attached files.

Reviewer #1: No

Reviewer #2: No

---

## [Author Response · Author response to Decision Letter 0]

24 Jul 2023

Thank you for the opportunity to revise and resubmit our manuscript. Below is a point-by-point response to the editor’s and reviewers’ comments. Changes made in the manuscript are bolded.

Regards

Fatma Shebl

Editor’s comments:

1. While everyone knows HIV, NHANES might be unfamiliar to those who do not live in the United States. Please write in full for first use, even in the Abstract.

Response: We added the full name as suggested.

Abstract, p.3

Objective: To investigate associations between all-cause mortality and human immunodeficiency virus (HIV) acquisition risk groups among people without HIV in the United States.

Methods: We used data from 23,657 National Health and Nutrition Examination Survey (NHANES) participants (2001-2014) and the Linked Mortality File...

2. What is the rationale for excluding women who have sex with women? 

Response: We excluded women who have sex with women (WSW) because the CDC does not include this population as one of the HIV acquisition risk groups; additionally, women who have sex with women are not in the comparator group for this study because they are not "heterosexually active.” We revised the text to clarify the rationale for the exclusion. 

 Methods, p.7-8

We also excluded women who have sex with women without a history of injection drug use (N=1,008) because they are not defined as an HIV acquisition risk group by the CDC [7] and are not "heterosexually active at low risk for HIV acquisition."

3. What paper/s served as the basis for your choice of covariates? 

Response: We added citations for the papers that served as a basis for the choice of covariates.

Methods, p.11

To examine mortality rates among individuals in the HIV acquisition risk groups, we assessed an a priori list of potential effect modifiers and confounders based on the published literature, including age, race/ethnicity, education, health insurance, health status, at-risk alcohol use, tobacco use, body mass index, condom use, lifetime sexual partners, and history of sexually transmitted infections [11, 12, 19-22].

4. Why was a chi-square test done instead of a log-rank test?

Response: We have generally used log-rank tests for the comparison of survival curves [1]. Since we were interested in comparing proportions, i.e., comparing the proportions of population characteristic variables between the risk groups, we used the chi-square test. We have clarified this in the manuscript methods.

Methods, p.11

For bivariate associations, we used the Rao-Scott F adjusted chi-square statistic to assess the relationship between covariates and the primary exposure (HIV acquisition risk groups) (i.e., to assess whether the distribution of the proportions of HIV acquisition risk groups is independent of the covariate) [25].

5. Justify the use of Cox regression over Poisson regression, considering that mortality rates are not expected to vary significantly during the time period.

Response: As noted by the reviewer, several options are available to model survival, including Cox, Poisson, and discrete-time logistic regression models [1, 2]. Historically, before the wide availability of Cox regression in statistical software, Cox models were approximated using Poisson regression [3]. Several studies have shown that the estimates are similar [3, 4]. Therefore, our decision regarding which model to use was based on two factors: 1) the Cox model is a semiparametric model; thus, we do not need to assume the baseline hazard, and 2) the availability of the statistical model in SAS. Given SAS does not include a procedure that employs Poisson regression in complex survey design but has procedures that can employ the Cox model or discrete-time logistic regression model in complex survey data, we decided to use the Cox model.

Methods, pp. 11-12

We assessed the relationship between covariates and the outcome of interest (all-cause mortality) using Cox proportional hazards regression models, accounting for the complex survey design [25]. Cox proportional hazards regression model is a semiparametric model; therefore, no assumption of the baseline hazard function is needed.

6. It is unclear what effect measure modifications were assessed and why they were done. What variables were considered as effect measure modifiers?

Response: We clarified that we assessed age as an effect modifier and why we selected this variable to assess as an effect measure modifier.

Methods, p.12

We then assessed whether "age at risk" is a potential effect modifier by including the HIV acquisition risk group, age, and the interaction term between age and HIV acquisition risk group in a Cox proportional hazards model. Age would be considered an effect modifier if the p-value of the interaction term was less than the alpha level of 0.05.

Results, p.16

We detected no significant interaction between age and HIV acquisition risk groups, F(6,104) = 1.88, p = 0.091 (S3 Table 3).

7. There had been issues with the change in estimate criterion in more recent published literature, which you have used in your paper. What is the justification for this choice of modeling strategy, and how will your model fare if you controlled for all known covariates regardless? I think your number of deaths is sufficient even if you controlled for all covariates which you have data for.

Response: We used the change in estimate criterion as a complementary method to evaluate a list of a priori potential confounders. It is advised not to control for variables that are not confounders, "unnecessary adjustment," since the adjustment can introduce bias [5]. Therefore, we elected not to adjust except for variables associated with both outcome and exposure that meaningfully changed the estimate. Although the dataset is large, we observed a sparse data problem because the outcome is relatively rare, which precluded us from including all the potential confounders in the model [6]. Our objective was to select the most parsimonious model; thus, we used both a priori list and the change in estimate criterion.

Discussion, pp.24-25

The use of the change in estimate approach to identify confounders has been criticized [41]. However, since it is advised not to control for variables that are not confounders to avoid introducing "unnecessary adjustment" bias [42], we used the change in estimate criterion as a complementary method to evaluate a list of a priori potential confounders. In identifying the list of a priori potential confounders, we followed the recommendations of VanderWeele 2019 [41] by selecting variables that could be causes of the exposure, outcome, or both, but excluding colliders, mediators, and instrumental variables. Although the dataset is large, we observed a sparse data problem because the outcome is relatively rare, which precluded us from including all the potential confounders in the model [43]. Therefore, we elected not to adjust except for variables associated with both outcome and exposure that meaningfully changed the estimate as determined by the change in estimate approach.

8. How was the proportional hazards assumption assessed? Please indicate in the methods. 

Response: The proportional hazards assumption assessment method was provided in supplemental appendix 1, and we have now added it to the main manuscript.

Methods, p.13

“Additionally, we assessed the Cox proportional hazard assumption using two standard approaches: 1) to visually assess the parallelisms of the lines of the graph of the log of -log survival versus the log of survival time (log-negative log plot), and 2) to include and assess the significance of the time-dependent variable in the model. If the lines were parallel and the time-dependent variable was not significant, then we considered the proportional hazard assumption to not be violated.”

9. Any results about your assessment of effect modification?

Response: We included the results of effect modification in the main text.

Results, p.16

We detected no significant interaction between age and HIV acquisition risk groups, F(6,104) = 1.88, p = 0.091 (S3 Table 3).

10. Line 284-288: "The attenuation of the magnitude of difference in mortality between male injection drug users and heterosexually active males at average risk for HIV after adjusting for confounders indicates that the excess mortality could be attributed to the health risk behaviors such as smoking and alcohol consumption, as well as racial disparity." I thought you have adjusted for smoking, alcohol consumption and race? What made you say these statements?

Response: We have adjusted for these variables, but residual confounding can exist after adjusting for covariates. Residual confounding can happen if the confounder is not perfectly measured; therefore, even after adjusting for the confounder, the effect of the confounder is not entirely removed. We have added a reference to this concept where we address this limitation.

Discussion p.24

Although we adjusted for multiple confounders, including smoking, alcohol use, health status, and race/ethnicity, residual confounding may persist due to missing data, measurement error, unreported confounders, or incomplete self-report especially because NHANES does not include data regarding time-varying covariates [40].

Reviewers’ comments:

Reviewer #1: The authors present a well-performed and well-written analysis of the NHANES datasets to assess the inherent risks of all-cause mortality in defined groups of HIV-negative persons at higher risk of HIV acquisition. The authors are asked to address the following comments:

1. Mortality was followed-up through 2015, but obviously more data might be available in the Linked Mortality File. Please explain and defend this end date for the analysis. If more data do exist, then the analyses should be redone to account for the longer follow-up period. 

Response: After this project's conception, analysis, and manuscript development, the linked mortality data then became available through 2019. Based on the reviewer's request, we revised the analysis to include the latest release of the mortality data and updated the text accordingly, including revising the methods. All results have been revised throughout the manuscript.

Methods, p.7

We used publicly available data from the 2001 to 2014 National Health and Nutrition Examination Survey (NHANES) [14] and the Linked Mortality File (LMF) [15], which has been updated with National Death Index (NDI) mortality follow-up data through December 31, 2019.

2. Please defend the exclusion of women who have sex with women. If they are not at higher risk for HIV acquisition, then why not include them in the analyses? 

Response: We excluded women who have sex with women (WSW) because the CDC does not include this population as one of the HIV acquisition risk groups; additionally, women who have sex with women are not in the comparator group for this study because they are not "heterosexually active.” We revised the text to clarify the rationale for this exclusion. 

Methods, p.7-8

“We also excluded women who had sex with women without a history of injection drug use (N=1,008) because they are not defined as an acquisition risk group by the CDC [7] and are not “heterosexually active at low risk for HIV acquisition.” 

3. Although the risk groups were stratified by sex, please clarify if there was a statistically significant interaction between risk group and sex on all-cause mortality.

Response: We revised the methods to clarify this.

Methods, p.12

We stratified risk groups by gender for two reasons, 1) there was a statistically significant interaction between the four-level risk group and gender on all-cause mortality, 2) it is more appropriate to compare women to women and men to men to account for unmeasured confounders that are related to self-reported gender, and the difference in mortality rates between women and men.

4. Please use 'sex' or 'sex at birth' instead of 'gender' throughout, unless gender is the truly assessed variable. 

Response: We used the term "gender" because the NHANES questionnaire used the following verbatim question: "Gender of the participant."

Methods, p. 10

“The remaining participants met the criteria for heterosexually active persons at average risk for HIV acquisition (HAH), stratified by self-reported gender as used in NHANES questionnaires.”

Reviewer #2: The authors should consider the following.

1. Unless certain that they are PWID it should be PWUD. 

Response: We used the terminology, “PWID,” because we defined this subgroup as people who self-reported in NHANES ever using a needle (Supplemental appendix 1).

 Methods, pp.8-9

“We categorized participants as “ever-PWID” if they self-reported ever using a needle to inject illegal or street drugs based on their answers to drug use questionnaires available for NHANES participants aged 20-59 years (Supplemental Methods). All other participants were classified as “never-PWID” or unknown PWID status based on their self-response to the PWID questions.”

2. Limitations should mention the time updated variables – e.g. antiretroviral therapy, viral load, nadir CD4 etc. 

Response: Because this population is restricted to people without HIV, data regarding antiretroviral therapy, viral load, and nadir CD4 are not available or relevant.

3. Figure 1 should be a table – it is not clear. 

Response: The data in Figure 1 require a very long table, which we include in the supplement and now reference in the main text.

Results, p.14

The associations between demographic characteristics and HIV acquisition risk groups are presented in Fig 1, S2 Results and S3 Table 1.

4. I would like the analyses repeated by race. 

Response: When we repeated the analysis by race, we ran into a sparse data bias issue because there were not enough case numbers for some of the combinations of the outcome and predictor, which was evident by very wide confidence intervals, unreasonable values, and convergence problems [6]. Special analytic methods are needed to address this issue, which is beyond the scope of this work. We have added this to the limitations.

Discussion, p. 24

“Given the small number of outcomes, we were not able to perform the analyses stratified by race.”

5. The discussion around the social determinants of health could be expanded. 

Response: We have expanded the discussion around social determinants of health to underscore that this is a critical issue to address among people at increased risk for HIV.

 Discussion, p.24

“A recently published analysis underscored that increasing numbers of disadvantageous social determinants of health were associated with all-cause mortality in the United States with a significant linear trend; among people with six or more disadvantageous social determinants of health, all-cause mortality was increased more than 7-fold [39]. Our findings highlight that heterosexually active people at increased risk for HIV acquisition also experience increased mortality, which further highlights the need to address social determinants of health and improve health outcomes among people with low income who are at risk for HIV acquisition.”

 

References

1. Riffenburgh RH. Chapter 23 - Survival, Logistic Regression, and Cox Regression. In: Riffenburgh RH, editor. Statistics in Medicine (Third Edition). San Diego: Academic Press; 2012. p. 491-508.

2. Suresh K, Severn C, Ghosh D. Survival prediction models: an introduction to discrete-time modeling. BMC Med Res Methodol. 2022;22(1):207. Epub 2022/07/27. doi: 10.1186/s12874-022-01679-6. PubMed PMID: 35883032; PubMed Central PMCID: PMCPMC9316420.

3. Dickman PW. Replicate a Cox model using Poisson regression 2019 [6/15/2023]. Available from: https://www.pauldickman.com/software/stata/compare-cox-poisson/.

4. Golden A, Ellis E, Le H. 0163 Comparison of risk estimates from cox proportional hazards and poisson modeling for association of occupational titanium dioxide exposure and selected causes of death. Occupational and Environmental Medicine. 2017;74(Suppl 1):A49. doi: 10.1136/oemed-2017-104636.131.

5. Schisterman EF, Cole SR, Platt RW. Overadjustment bias and unnecessary adjustment in epidemiologic studies. Epidemiology. 2009;20(4):488-95. Epub 2009/06/16. doi: 10.1097/EDE.0b013e3181a819a1. PubMed PMID: 19525685; PubMed Central PMCID: PMCPMC2744485.

6. Greenland S, Mansournia MA, Altman DG. Sparse data bias: a problem hiding in plain sight. BMJ. 2016;352:i1981. Epub 2016/04/29. doi: 10.1136/bmj.i1981. PubMed PMID: 27121591.

---

## [Decision Letter · Decision Letter 1]

2 Aug 2023

The association between all-cause mortality and HIV acquisition risk groups in the United States, 2001-2014

PONE-D-23-05219R1

Dear Dr. Shebl,

We’re pleased to inform you that your manuscript has been judged scientifically suitable for publication and will be formally accepted for publication once it meets all outstanding technical requirements.

Kind regards,

Veincent Christian Pepito

Academic Editor

PLOS ONE

Additional Editor Comments (optional):

Thank you very much and congratulations on your newly accepted paper. Please don't forget to spell out NHANES in full in your Abstract.

Reviewers' comments:

Reviewer's Responses to Questions

**Comments to the Author**

1. If the authors have adequately addressed your comments raised in a previous round of review and you feel that this manuscript is now acceptable for publication, you may indicate that here to bypass the “Comments to the Author” section, enter your conflict of interest statement in the “Confidential to Editor” section, and submit your "Accept" recommendation.

Reviewer #1: All comments have been addressed

Reviewer #2: All comments have been addressed

2. Is the manuscript technically sound, and do the data support the conclusions?

Reviewer #1: Yes

Reviewer #2: Yes

3. Has the statistical analysis been performed appropriately and rigorously? 

Reviewer #1: Yes

Reviewer #2: Yes

4. Have the authors made all data underlying the findings in their manuscript fully available?

Reviewer #1: Yes

Reviewer #2: Yes

5. Is the manuscript presented in an intelligible fashion and written in standard English?

Reviewer #1: Yes

Reviewer #2: Yes

6. Review Comments to the Author

Reviewer #1: (No Response)

Reviewer #2: (No Response)

7. PLOS authors have the option to publish the peer review history of their article (what does this mean?). If published, this will include your full peer review and any attached files.

Reviewer #1: No

Reviewer #2: No

---

## [Editor Report · Acceptance letter]

8 Aug 2023

PONE-D-23-05219R1 

The association between all-cause mortality and HIV acquisition risk groups in the United States, 2001-2014 

Dear Dr. Shebl:

I'm pleased to inform you that your manuscript has been deemed suitable for publication in PLOS ONE. Congratulations! Your manuscript is now with our production department. 

Kind regards, 

on behalf of

Mr Veincent Christian Pepito 

Academic Editor

PLOS ONE